# Cytocompatibility and Bioactive Ion Release Profiles of Phosphoserine Bone Adhesive: Bridge from In Vitro to In Vivo

**DOI:** 10.3390/biomedicines10040736

**Published:** 2022-03-22

**Authors:** Kateřina Vrchovecká, Monika Pávková-Goldbergová, Håkan Engqvist, Michael Pujari-Palmer

**Affiliations:** 1Department of Pathology Physiology, Faculty of Medicine, Masaryk University, 62500 Brno, Czech Republic; vrchoveckak@mail.muni.cz (K.V.); goldberg@med.muni.cz (M.P.-G.); 2Department of Materials Science and Engineering, Applied Material Science, Uppsala University, 75103 Uppsala, Sweden

**Keywords:** phosphoserine, bone tissue adhesive, calcium phosphate cement, odontoblast, osteoblast, cytotoxicity, ion release, in vitro

## Abstract

One major challenge when developing new biomaterials is translating in vitro testing to in vivo models. We have recently shown that a single formulation of a bone tissue adhesive, phosphoserine modified cement (PMC), is safe and resorbable in vivo. Herein, we screened many new adhesive formulations, for cytocompatibility and bioactive ion release, with three cell lines: MDPC23 odontoblasts, MC3T3 preosteoblasts, and L929 fibroblasts. Most formulations were cytocompatible by indirect contact testing (ISO 10993-12). Formulations with larger amounts of phosphoserine (>50%) had delayed setting times, greater ion release, and cytotoxicity in vitro. The trends in ion release from the adhesive that were cured for 24 h (standard for in vitro) were similar to release from the adhesives cured only for 5–10 min (standard for in vivo), suggesting that we may be able to predict the material behavior in vivo, using in vitro methods. Adhesives containing calcium phosphate and silicate were both cytocompatible for seven days in direct contact with cell monolayers, and ion release increased the alkaline phosphatase (ALP) activity in odontoblasts, but not pre-osteoblasts. This is the first study evaluating how PMC formulation affects osteogenic cell differentiation (ALP), cytocompatibility, and ion release, using in situ curing conditions similar to conditions in vivo.

## 1. Introduction

There is growing interest in biomaterials to replace or augment tissues that have difficulty healing after injury or disease. Since the discovery of the self-setting cement [1], calcium phosphate cement (CPC) has become the most common biomaterial for the reconstruction of bone-related defects due to its favorable biocompatibility, osteoconduction, osteointegration, and chemical similarity to the inorganic part of the bone (hydroxyapatite) [2]. CPCs are used to treat bone trauma and fractures, replace damaged tissue, or fill bone cavities, and are also commonly used in dentistry [3,4,5,6,7]. Another advantage is the rapid formation of a physical bond between the bone and the graft [3]; ease of handling, which reduces the time required for surgery; and minimal postoperative trauma [4,5,8].

CPCs remain a synthetic material, with physical properties that are inferior to native tissue [9] and lack a hierarchal organization, organic and inorganic phase integration, or biomineralization [5,10]. The most important limitation for CPCs appears to be their low fracture toughness, moderate mechanical strength, brittleness, low impact resistance, and low tensile stress compared to bone tissue. The main source of these limitations is material brittleness and porosity, which serve as the initiation sites for crack propagation [11]. This limits the use of CPCs to primarily non-load-bearing applications (e.g., middle ear surgery), “void fillers” (e.g., bone fracture and defect), or “coatings” (e.g., dental and orthopedic metallic implants). The brittleness arises from the type of chemical/physical bonds (e.g., ionic bonding), which do not re-form after cracks arise [12]. A popular approach to improve the biological and physical (mechanical) properties is additives (e.g., amino acids, polymers, chelators of calcium, organic acid, and elemental/ionic doping), which can play complimentary roles in bone metabolism [2,13]. Modified cements can provide short-term biologically desirable properties while also being resorbed and replaced by new bone. Effective biomaterials should allow the bone defect to be gradually and physiologically healed without compromising the stability of the affected bone segment [14].

One of the most promising additives is phosphoserine (PSer). The first reports of phosphorylated amino acids as additives indicated that phosphoserine improved the physical and biological properties of cement [5,14]. PSer is a necessary component (i.e., dephosphorylation causes loss of function) of many proteins in the bone matrix (e.g., osteopontin), plays an important role in bone resorption, and is responsible for interactions between the organic bone matrix proteins and inorganic phases like hydroxyapatite, underpinning load-bearing fracture toughness and load dissipation [15,16]. The high affinity of PSer carboxyl groups for calcium phosphate leads to the inhibition of HA crystal growth and the formation of nanocrystals with a higher specific area [17]. PSer also facilitates the migration and adherence of macrophages to implants, which play an important role in material resorption [18] and promote osteoblast proliferation and differentiation [4,19,20]. Additionally, PSer has positive effects on the biomechanical properties of calcium cements, accelerating setting times and increasing the mechanical strength [21]. More recently, PSer has been shown to react with certain calcium phosphate phases to form a biocompatible organo-ceramic self-curing cement with adhesive properties (up to 40 times stronger adhesion to the tissues and biomaterial surfaces than unmodified cement) [5,22]. The addition of Pser to α-TCP creates a material with more similar architecture and physical properties to native calcified bone [6,11,23]. 

Ex vivo studies have shown faster remodeling of phosphoserine modified cements (PMCs) compared to unmodified aTCP cement, where modified cements rapidly transform into brushite rather than slowly transforming into hydroxyapatite [4,5]. In vivo studies have shown osteointegration and histocompatibility of phosphoserine cements [4,17,21,24], where the tissue healing response is improved without pathological inflammation, and is remodelled into new bone rapidly [5]. Early studies on PMCs used cements with much lower concentrations of phosphoserine (1–5 wt%), and CPCs with phosphoserine stimulated higher bone remodeling and bone formation in rats [18] and accelerated the resorption of calcium phosphate cements in mini pigs [17]. However, there have been no in vitro studies evaluating the exact cellular mechanisms underlying the cell-level and tissue-level changes that occur during healing, which are unique to phosphoserine cements. 

Understanding the biological interactions between biomaterials and tissues is an important aspect in evaluating the stability of the implant. Immediately after a bone injury, blood- and tissue-derived signals (e.g., cytokines) recruit (a) immune cells (e.g., macrophages), and polarize them to secrete tissue regeneration (M2) and/or inflammation signals (M1); (b) fibroblasts, which create the extracellular matrix; (c) angiogenic cells (e.g., endothelial cells), to create the necessary blood vessels to feed newly formed bone; (d) osteogenic cells (e.g., mesenchymal stromal and stem cells (MSC), osteoblasts, chondrocytes), and differentiate them to form the callus and create new bone mineralization; and (e) mineral and matrix resorbing cells (e.g., osteoclasts), to convert the callus into new woven bone. Soluble factors, including ions, released during the earliest time points following injury are critical to defining the subsequent inflammation and healing response [25,26]. The “polarization” of immune cells and ion (calcium) concentrations [27] directly define the subsequent cytokine release and recruitment/activation of the osteogenic and angiogenic process. During the healing process, calcium-based biomaterials, including most bone void fillers and PMCs, influence the majority of the cells acting in healing on each of these cell types through a combination of mechanisms: material surface properties [28,29], ion release [30], porosity [31], etc. As little is known about how PMCs behave in direct contact with tissues [4], and about the degradation and subsequent changes in surface properties [5], this study focused on indirect contact effects that can influence each stage of healing—ion release. 

Only one study has investigated how cells react to PMCs [4], and there is no information on how the composition of PMCs affects the cells involved in bone regeneration (e.g., osteoblasts, osteoclasts, and MSCs). For example, in the present study, the PMC composition was modified to include partial or complete replacement of calcium phosphate with calcium silicate (pseudowollastonite), which can be a potent stimulator of bone tissue healing (vol/wt 2–28%) and development [32]. Moreover, calcium silicate is essential for bone cell activity [4,8], promotes osteoblast proliferation and differentiation [32], and increases fracture toughness and compressive strength [32]. With the exception of a few recent studies [30,33], it is known, but rarely discussed, that in situ setting calcium phosphates are well tolerated in vivo, but are often cytotoxic in vitro [34]. The amount of ion release can easily exceed the buffering capacity of static cultures within minutes, even when CPCs are cured for hours to days [3,35]. Dos Santos et al. and Miyamoto et al. reported a similar phenomenon using calcium phosphate cement with slow setting times in subcutaneous rat tissue, demonstrating that the setting reaction must take place in order to preclude an excessive inflammatory response [35]. The disconnect between in vitro and in vivo testing conditions is a primary contributor to the poor correlation between in vitro and in vivo studies on bioceramics [36]. The purpose of this study was to evaluate how different formulations of PMC affected (A) ion release profiles and (B) cytotoxicity in multiple cell lines (MDPC 23, L929, and MC3T3), and (C) to try to “bridge” in vitro and in vivo study methods. We evaluated how different handling conditions, curing times, sample dimensions, formulations, etc., affect curing and ion release. We also evaluated cytotoxicity to identify how formulation and composition affects the safety of PMCs (ISO 10993). Finally, we tried to modify typical test methods (e.g., shifting from static to dynamic culture) to more closely represent in vivo conditions. In the present study, we used rapidly curing formulations in-situ, and attempted to overcome the limitations of static, mono-cell-type cultures, and to more accurately mimic the initial cell−material interactions that occur in vivo.

## 2. Materials and Methods

### 2.1. Materials 

All of the materials were purchased from Sigma-Aldrich (AB Sigma-Aldrich, Stockholm, Sweden), unless otherwise indicated. Alpha tricalcium phosphate (αTCP, Ca_3_(PO_4_)_2_) was synthesized by heating calcium carbonate and monocalcium phosphate anhydrous, at a 2:1 molar ratio, on a zirconia setter plate for 12 h, at 1450 °C in a furnace (Entech MF 4-16 chamber furnace with silicon carbide heating element and Eurotherm 3208 controller). After quenching in air, the αTCP powder was dry milled (Reitsch PM400, AB Ninolab, Stockholm, Sweden) in a 500 mL zirconia milling jar, at 300 RPM for 15 min, with 100 g of powder per 100 zirconia milling balls (10 mm diameter), and the purity was determined to be 98% wt% αTCP, with 2 wt% αTCP as an impurity, by Rietveld refinement of X-ray diffraction patterns (Profex software). O-phospho-L-serine, referred to hereafter as phosphoserine, was purchased from Flamma SpA (>95%, Flamma SpA, Bergamo, Italy). Calcium metasilicate (psueodwollastonite, CS1, 99%) was purchased from Sigma Aldrich and was used as received. 

### 2.2. Sample Preparation 

Samples were created by hand-mixing powders, as described previously [5], with a defined mole percentage of phosphoserine (Pser) to αTCP and/or calcium metasilicate (CS1, Pseudwollastonite) (Appendix A). Deionized water was added as the liquid (0.25 mL per gram, liquid to powder (L/P) ratio for all samples), and the samples were hand mixed with a spatula for 10 s. Disc samples were cast in silicon molds (8.9 mm diameter × 4.5 cm thickness), with a fixed sample size of 0.375 g, and were allowed to cure for 5 min or the initial setting time before incubating in cultivation media. Cylinder samples were cast in Teflon molds (1.6 mm diameter × 1.6 cm thickness), prepared from 0.100 g of each sample formulation, and allowed to cure for an initial setting time before incubating in the cultivation media.

Samples for setting time testing, pH, and cytotoxicity testing (indirect contact) were made from 0.375 g of powder and 0.09375 mL liquid (L/P of 0.25), molded into 8.9 mm × 4.5 mm discs, allowed to cure for 5 min (preliminary testing) or until the initial setting time (Table 1) for each formulation, weighed, and incubated in media without antibiotics or serum at 0.2 g mL^−1^, according to ISO 10993-12 and 10993-5, in 37 °C in 5% CO_2_ atmosphere in a 24-well tissue culture plate. Samples for ICP and ALP testing were fabricated in 1.6 mm × 1.6 mm cylinder molds for the initial setting time, and were then moved to 24-well plates containing cells (ALP) or without cells (ICP).

### 2.3. Material Characterization 

X-ray diffraction (XRD): The properties of precursors αTCP, phosphoserine, and calcium metasilicate (pseudowollastonite) used in the present study have previously been described [5]. X-ray diffractograms (XRD) of the powder samples were obtained on a Bruker D800 Advance (Bruker Daltonics Scandinavia AB, Solna, Sweden), and were scanned with a step size of 0.03 degrees per step and a dwell time of 0.2 s per step, from 3 to 60 degrees. The phase composition was identified by Rietveld refinement using reference files: PDF# 04-010-4348 alpha tricalcium phosphate (αTCP), #04-008-8714 beta-tricalcium phosphate (βTCP), #01-074-0565 hydroxyapatite (HA), #04-011-1625 tetracalcium phosphate (TTCP), #04-013-3344 dicalcium phosphate dihydrate (DCPD), #04013-3883 octacalcium phosphate (OCP), and #04-007-9734 CaO. All crystalline peaks were accounted for, and any phases comprising less than 1% (below limit of detection of approximately 1–2% weight %) were removed before the final Rietveld analysis. 

Fourier transformed infrared spectroscopy (FTIR): The precipitated crystals that formed over 24-h incubation in media, as described in the cytotoxicity testing below, were collected by rinsing (distilled water) and drying (ethanol rinse), and the collected powder was analyzed with a Bruker Tensor 27 (Bruker Daltonics Scandinavia AB, Solna, Sweden), with a platinum ATR attachment. Absorbance spectra were collected over a range of 400–4000 cm^−1^. 

Setting time: Setting times of the cement samples were measured using the Gilmore needle, according to the ASTM C266-04 standard. The initial setting time needle had a tip diameter of 2.12 ± 0.05 mm and a mass of 113.4 ± 0.5 g. The final setting time needle had a tip diameter of 1.06 ± 0.05 mm and a mass of 453.6 ± 0.5 g. Each measurement was repeated five times, and an average value was calculated, using a disc-shaped mold with an 8.9 mm diameter × 4.5 cm thickness. To measure the setting time, needles were gently lowered into the surface of each disc and were allowed to rest for 3 s before removing the needle. The setting time endpoint was declared when the needle no longer left a complete circular mark on the specimen surface. All experiments were performed at room temperature (21 ± 1 °C) and were cured in a sealed, humidified plastic container. 

### 2.4. pH Neutralization of Samples 

The media containing extracts from sample discs were collected in 15 mL tubes, and the volume was recorded and pH was measured (Seven compact pH meter, Mettler-Toledo AB, Stockholm, Sweden), and they were neutralized using 10 M NaOH when the pH was below 7, or 1 M NaOH between 7 to 7.4 pH. The new volume was then recorded, and the liquid was sterilized via filtration (0.2 µm PTFE filter, Thermofisher Scientific AB, Stockholm, Sweden) and used immediately or stored at −20 °C for later use. 

### 2.5. ICP-OES 

Ion concentration was evaluated using inductively coupled plasma optical emission spectrometry (ICP-OES, Avio 200, Perkin-Elmer spectrometer). Two sizes of samples were used: discs with a diameter 8.9 mm and thickness of 4.5 mm (identical to cytotoxicity testing samples) or cylinders with a diameter 1.6 mm and thickness of 1.6 mm. The samples were incubated in a cultivation media (DMEM/F12) without serum or antibiotics, at 37 °C, in 5% CO_2_, at 0.2 g of sample per mL of media for discs, or 0.015 g of sample per 2 mL of media (cylinders), with ion release beginning at the initial setting time (samples not incubated in media until the initial setting time was reached). During preliminary testing, the following ions were evaluated: Ca, Na, Si, Mg, P, Zn, Sr, Fe, Zr, Mn, Al, and S. We used a standard solution ICP-OES for Ca, Mg, Sr, Zn, Zr, Al, S, P, Na, Mn, Si, and Fe. Firstly, we did a calibration line (0–1000 mg/L). Only the following ions were found in significant quantities therefore the final testing focused on Ca, Na, Si, Mg, P, Zn, and Sr ions, with complete media replacement and collection occurring after 1 h, 2 h, 4 h, 6 h, 24 h, 32 h, 48 h, 72 h, and 5 days of samples incubation. Next, we did only two-point calibration before each measurement: Ca 0–10 mg/L, Mg 0–1 mg/L, Mn 0–0.1 mg/L, Al 0–0.1 mg/L, Sr 0–0.1 mg/L, Na 0–100 mg/L, Zn 0–0.1 mg/L, Fe 0-0.1 mg/L, Zr 0–0.1 mg/L, Si 0–10 mg/L, P 0–10 mg/L, and S 0–50 mg/L. Each measurement was repeated three times on four independent samples. The results are presented as mean value ± standard deviation. 

### 2.6. Cell Culture 

The mouse odontoblast-like cell line (MDPC 23, a gift from Jacques Nor at University of Michigan) and mouse fibroblast (L929, American type culture collection, ATCC) were cultured in Dulbecco’s modified Eagles medium/F12 (Gibco, New York City, NY, USA), supplemented with 10% fetal bovine serum (FBS, HyClone, Logan, UT, USA) and 1% penicillin/streptomycin (SigmaAldrich, St. Louis, MO, USA), at 37 °C in a 5% CO_2_ atmosphere. 

The pre-osteoblast cell line (MC3T3 E14, ATCC CRL2594, LGC Standards GmbH, Wesel, Germany) was expanded in alpha modified Eagles medium (αMEM Gibco, New New York City, NY, USA), lacking ascorbic acid and supplemented with 10% fetal bovine serum (FBS, HyClone) and 1% penicillin/streptomycin (Sigma-Aldrich, St. Louis, MO, USA) at 37 °C in a 5% CO_2_ atmosphere. For differentiation studies, MC3T3 were cultured in αMEM containing ascorbic acid (αMEM Hyclone, Logan, UT, USA). The cell morphology of the morphology was observed by an inverse phase-contrast microscope (Olympus) at 10× magnification. 

### 2.7. Cytotoxicity of Cement Extracts 

MDPC23, MC3T3, and L929 cells were seeded at a density of 6 × 10^3^ cells per cm^2^ on a 96-well plate and were cultivated for 48 h until reaching approximately 60% confluences. Cement extracts of all of the investigated samples were prepared in accordance with ISO 10993-12 and 10993-5. Disc samples (8.9 mm diameter and 4.5 mm thickness) were cured until the initial setting time was reached and were immersed in cultivation media (0.2 g/mL ratio) for 24 h at 37 °C in a 5% CO_2_ atmosphere. After 24 h, the extracts were collected; the pH of the extracts were measured, neutralized, and sterilized by 0.2 µm PTFE filter; and supplements (10% FBS, 1% pen/strep) were added. Media in 96-well plates were completely replaced with undiluted extracts (100%), and the cells were cultured for 48 h. The cell metabolic activity was measured by Alamar. 

Blue assay (Invitrogen Thermofisher Scientific AB, Stockholm, Sweden), using a 5% solution of Alamar blue, was diluted into media containing 10% FBS and 1% pen/strep. Cells were incubated for 1 h, and 100 µL of media were transferred to a clear plastic tissue culture plate, and fluorescence was measured at 560 nm and 590 nm on a spectrophotometer plate reader (excitation, emission, Infinite M200, Tekan, Switzerland). The blank (5% Alamar blue solution) was subtracted from each sample, and the optical density was normalized to the untreated control in order to obtain a percentage representing the “survival” [37]. 

### 2.8. ALP Activity and MicroBCA 

MDPC 23 and MC3T3 cells were seeded at a density of 10,000 cells cm^−2^ or 5000 cells cm^−2^, on a 24-well plate, respectively, and allowed to reach confluence. The media were replaced, and 1.6 × 1.6 mm cement cylinders were placed onto the cell monolayer with 2 mL differentiation media (50 μg/mL ascorbic acid, 10 mM β-glycerol phosphate. After 3 or 7 days, the media were removed, and 300 µL of Cell-Lytic (Sigma Aldrich, St. Louis, MO, USA) was added, and the samples were immediately frozen at −20 °C for later analysis. 

A 25 µL aliquot of MC3T3 lysate, or 2 µL of MDPC 23 lysate, were taken from each well and combined with 50 µL of alkaline phosphatase substrate for 4 min at 37 °C (total volume 75 µL). The reaction was stopped by adding 25 µL of 3 M NaOH. The absorbance was measured on a Tekan plate reader at 405 nm. For the BCA analysis, a 50 µL aliquot was taken from each sample lysate and was combined with an equal volume of Micro-BCA solution (Thermofisher Scientific AB, Stockholm, Sweden) and incubated at 37 °C for 1 h. The absorbance was measured on a Tekan plate reader at 562 nm. The resulting ALP, BCA, and ALP/BCA values were calculated as previously described [38]. 

### 2.9. Statistical Analysis 

Statistically significant differences were identified with SPSS software (version 22), with one-way ANOVA. Details are provided in the captions for each figure; briefly, in Figure 5, for each ion and at each time point, the cumulative ion concentration at 24 h is compared between “static” and “dynamic” samples for the same group (e.g., #38 static vs. #38 dynamic) using the Games−Howell post hoc test, and for each time point of dynamic samples, the cumulative ion release is compared between all groups and the “control” group, which released the most ions (#38), using the Dunnets post hoc test. In Figure 6, the mean cumulative ion concentration at each time point, for each ion, is compared between different sized samples using the Games−Howell post hoc test. In Figure 7, for each time point, the cumulative ion release is compared between all group means and the “control” group, which released the most ions (#34), using the Games−Howell post hoc test. In Figure 8, the viability of each treated cell group is compared to the mean values for control (untreated) using Dunnets (Figure 8B), or between all groups and the control (untreated) using the Games−Howell post hoc test. In Figure 9, the total protein and ALP are each compared between all treatment groups and control (untreated) means, for each cell line and condition, using the Games−Howell post hoc test. Statistical analysis was run on groups with samples sizes of three per group for ICP-OES testing, six per group for viability testing, and four per group for ALP and protein levels.

## 3. Results

### 3.1. Adhesive Formulations That Produce pH Neutral Extracts, Indirect Contact 

A series of adhesives (PMCs) were synthesized by varying the molar ratio of the amino acid and calcium salts, αTCP, and/or CS1 (calcium metasilicate, pseudowollastonite), with the aim of finding pH neutral and cytocompatible formulations for testing in a static cell culture. A total of 54 different formulations were screened, and 17 formulations were selected for analysis in vitro (an abbreviated list of the most promising formulations is shown in Table 1, while Appendix A lists all of the evaluated formulations). The initial screening criterion was neutral pH extracts (between pH 6–8) and rapid initial setting times (2–5 min). Formulations #29, #34, and #35 were also included because they were evaluated in prior or ongoing in vivo studies [4,5], and have a high bond strength to calcified or soft tissues [5,24]. 

The first challenge was to determine the most appropriate test conditions for evaluating the cytotoxicity of an in-situ curing tissue adhesive, in vitro (Figure 1). The standard methods (ISO 10993) for evaluating the cytotoxicity of biomaterials (indirect contact) are designed for static materials, while PMCs change dramatically, and are intended for use during the setting process. We sought to identify a compromise between standard static cytotoxicity testing, where the materials have finished curing and ion release is minimal, and conditions are more similar to in vivo use, where significant ion and amino acid release occurs, leading to supersaturation during the in-situ setting reaction. 

Briefly, the approach taken in this study was to (a) collect leach from in-situ curing PMCs for indirect testing (soluble fraction only, no precipitants); (b) screen the ion concentrations released in the media using ICP-OES; (c) compare how ion release (soluble fraction only) was affected by the composition, sample dimensions (e.g., mass, shape), curing time, and incubation time; and (d) how indirect contact and direct contact with the adhesive affected the survival and differentiation of osteogenic cell types: odontoblasts (MDPC-23) and osteoblasts (MC3T3-E1) (Figure 2).

When attempting to compare the extract pH and ion release between different PMC formulations, it became clear that using an arbitrary setting time for all groups (e.g., cure for 24 h before testing) was unacceptable because each composition had significantly different (Gilmore needle) setting times (Figure 3A). Using only fully cured samples (>24 h) for experiments would neglect the significant differences in the material properties that arise during the in situ curing phase (first 10 min). As the clinical requirement for in-situ curing cements and adhesives typically includes a working time of 5–10 min [39], we first compared how the setting time affected the pH of the extracts (incubated in media for 24 h at 37 °C, 200 mg mL^−1^) by curing a range of formulations for either a fixed time (5 min, Figure 3B) or until the initial setting time (Figure 3C). 

The curing time affected the composition range where neutral extracts were obtained: when cured for 5 min, neutral (Figure 3B) pH extracts (7.4 ± 0.2) were obtained from silicate containing PMCs with 25–35 mole% phosphoserine. In contrast, when cured to the initial setting time, the average pH over this same composition range was higher by 0.2–0.4 pH, suggesting that the amount of released ions, or the ratios between Ca, P and Si, were altered by prolonging the curing time. When cured for 5 min (Figure 3B), PMCs containing 30% phosphoserine or less, and at least 50% of the calcium salt as silicate (pseudowollastonite), produced neutral pH extracts, after immersion in media for 24 h. A higher phosphoserine or lower silicate content produced acidic extracts. Formulations that produced pH neutral media after 24 h are colored in black. The plots in Figure 3 indicate that adhesive formulations in the range of 25–35 mole% phosphoserine, with at least 50 mole% aTCP, exhibited both a rapid setting time and pH neutral. These extracts are the most likely to be non-cytotoxic. All subsequent testing used samples cured to the initial setting time before testing. 

Phosphoserine (Pser) retarded the setting time when it comprised >50 mole%, while CS1 accelerated the setting time (Figure 3A). PMCs formulations containing <35 mole% phosphoserine cured within 3 min, 35–50 mole% phosphoserine cured between 3 and 27 min, and >50 mole% phosphoserine PMCs, required longer than 27 min to reach the initial setting time. Testing was restricted to the range of 15–80 mole % Pser, as outside these ranges the adhesive did not set. Phosphoserine is a strong acid (5.6 pKa value [40]), aTCP is weakly alkaline (7.9 pH in medium [41])**,** and calcium metasilicate (psueodwollastonite) is very alkaline (9.9 pH value [42]). Differences in the pH values, between reactants in acid-based reactions, including cementitious reactions, produce both chemical energy and heat, thereby accelerating the reaction. Increasing quantities of calcium metasilicate accelerated the setting time. A similar accelerated setting effect was observed when tricalcium phosphate was replaced with a more basic calcium phosphate salt, tetracalcium phosphate, in separate studies (data not shown), indicating that the accelerated setting time was likely due to pH effects from the calcium salt. 

Of the formulations tested in Figure 3 and Table 1, only four formulations were found to produce neutral extracts (pH 7.4 ± 0.2) after incubation for 24 h at 0.2 g mL^−1^ of media (#28, #32, #34, and #39). In vitro systems lack the buffering capacity of tissues. In vitro testing of acidic cements has, until now, been largely impossible without curing samples for days to weeks, as well as repeated rinsing to deplete excess ions [33,43]. PMCs with extract pH as low as 5.7 and 2.8 (#29 [4] and #36 [4], respectively, and #35 [24]) are histocompatible in vivo [4], while, in vitro, these formulations would likely be cytotoxic due to pH and ion related effects. Most importantly, regardless of the pH or whether extracts were cytocompatible, most PMCs discs produced a thick carpet of precipitated crystals within 6 h of incubation in media. Therefore, collecting the soluble extracts and measuring the pH and cytocompatibility, via indirect contact, gives an incomplete picture of cytocompatibility, due, in part, to the limitations of the in vitro culture in reproducing in vivo conditions (e.g., rapid replenishment of fluids in vivo). When PMCs were cured for 24 h or longer prior to immersion in media, no crystals precipitated. The present results strongly reflect the chosen culture conditions (Figure 3A–C); using larger volumes of media, differing frequency of media replacement, curing time, or any number of other factors, will change the outcomes reported here. The present experiments were designed to study ion release in near-physiological fluids, during the in-situ curing phase (first 24 h). The soluble fraction of ions, which are capable of diffusing to cells and tissues to stimulate gene expression changes in vivo, were the focus of this study. 

### 3.2. Precipitants and Characterization (FTIR/XRD/IMAGES) 

Prior to characterizing the concentration of ions released from each PMCs, we attempted to characterize the insoluble precipitants, for a selected group of PMCs formulations. When the ionic concentration in solution approaches supersaturation, crystalline phases will nucleate, followed by crystal growth [44]. The relative magnitude of precipitant formation (qualitative) is shown for four formulations in Figure 4. A small amount of crystals appeared within 2–6 h, with significant precipitation occurring between 6 and 24 h. Incubation volumes greater than 100 mL (equivalent to 5 L of fluid per gram of adhesive) per sample were needed for “discs” (8.6 × 4.5 mm) and “cylinders” (1.6 × 1.6 mm) to prevent precipitants from forming (e.g., approaching perfect “sink” conditions). The morphology of precipitants appeared spherical, including individual submicron sized plates, needles, spheres, and spherical agglomerations composed of plates or needles (Figure 4F). Electron microscopy analysis was not used as the dehydration process would likely alter the microstructure, as some precipitants appeared to include amorphous hydrates. None of the expected crystalline calcium phosphate or calcium silicate phases were detected (e.g., brushite; apatite; meta-, di-, or tri- calcium silicate) by XRD (Figure 4D) or FTIR-AR (Figure 4E). Instead, the closest IR and diffraction matches (software and database) appeared to be mixed salts of calcium, magnesium, and silicate. 

### 3.3. Ion Release 

#### 3.3.1. Ion Release Time Course (Cumulative) and Effect of Media Replenishment 

Ion release from PMCs was evaluated under two conditions: 24 h incubation without media replenishment (“static”), identical to the conditions used for cytotoxicity and pH testing, or repeated media replenishment to more closely simulate in vivo conditions (“dynamic”) (Figure 5). After screening the ion release of 17 different formulations (Appendix A) and comparing viability (preliminary data not shown), four formulations were selected that represented a range of compositions and viability, and these four were used for all subsequent testing: #29, #35, #38, and #39. The preliminary data also confirmed that other contaminating ions, which are common in many calcium phosphates and silicates, were not present in significant quantities: Zn, Sr, Fe, Al, Zr, Na, and Mn. Preliminary testing also confirmed that (Al, Fe, Mn, S, and Zr) were not found in different PMCs formulations. Briefly, #29 and #35 have been used previously in vivo and are known to be “safe” [4,45]. Formulations #29 and #39 contained identical phosphoserine concentrations, but varied in amounts of silicate; #35 contained no silicate, and #38 was cytotoxic in vitro (acting as a “positive internal control” for later cytotoxicity testing). 

The relative rate of ion release was proportional to the amount of Pser and the setting time, and inversely proportional to the amount of CS1. The highest rate of cumulative ion release occurred in the formulation with the longest cure time and highest amount of Pser, #38 (54 mole% Pser) (Figure 5A–C). The same data are presented using the non-cumulative release rate in Appendix A. When comparing between groups at each time point, the research question was whether each formulation produced significantly less ions than the highest releasing group (Group 38 for Figure 5) at each time point. 

In the first hour, #38 released two-fold more calcium, three-fold more phosphorus, and between two- and thirty-fold more silicon than #29, #35, or #39 (Figure 5C). This pattern was largely reversed by 24 h through to 5 days, where #29, #35, and #39 released 2.5–3 fold more calcium, and 1.5-fold more phosphorous than #38. The highest ion release occurred in the first hour for all formulations, with 30–50-fold higher concentrations of ion released compared to the DMEM control media, for calcium and phosphorus ions, and 15–25-fold higher concentrations of silicon. In the first hour of release, a range of 20–45 mM of calcium, 40–120 mM phosphorus, and 3–10 mM of silicon, was found in the extract media. Chronic ion release (1–5 days) concentrations are showed in Table 2. 

For reference, the concentrations above which precipitants are reported to form in DMEM media, for calcium, phosphorus, and silicon, are <0.17–5 mM, <2 mM [46], and <1 mM [47], respectively. After the first day, ion release remained low through to 5 days. This trend is shown in Table 3. 

The ion release profiles were also compared between dynamic and static conditions, with static conditions representing the typical ISO-10993 method of evaluating cytotoxicity (“static” data points in Figure 5A–C) after 24 h. Figure 4 is presented as cumulative release to allow for a direct comparison between static and dynamic ion release at 24 h. Unsurprisingly, replenishment of media led to a 2.5–4 fold greater cumulative release of calcium and phosphorus after 1 day, and up to 7-fold more silicon, compared to the static conditions for each formulation. Most importantly, the relative trend of ion release remained the same, regardless of test conditions: #38 released more calcium, phosphorus, and silicon than other formulations; #29 and #39 released the second highest levels of calcium; #29 and #35 released the second highest levels of phosphorus; and #39 and #29 released the second highest levels of silicon (*p* > 0.05, Games−Howell). 

These results suggest that (a) the short term trends (up to 5 days) shown in Figure 4 are likely to accurately predict differences in the relative rates of ion release, between different formulations, both in vitro and in vivo; (b) even after exceeding solubility, static and dynamic samples exhibited similar trends of ion elution; and (c) if cytocompatibility and cellular testing only use extracts collected within the first 24 h, important changes in ion release that will occur in vivo will be neglected (e.g., #38 releases 2-fold more calcium in the first hour, and 2-fold less calcium throughout days 1 to 5, compared to #29, #35, and #39). Since the focus of this study was the soluble fraction of ions, precipitants were not dissolved or measured. The amount of released ions that remained, as precipitation, represent an important missing part of the true ion release rate, at present. 

A correlation model shows the relationship between setting time, pH, composition and ion release rates for each ion. The amount of calcium released (Figure 5A) decreased with decreasing the PSer content in the formula (#38—#29—#39—#35), and decreased with faster setting times, regardless of the further proportion of components in the material. The lowest phosphorus concentration released was measured in formula #39, which contained the least amount of calcium salt (α-TCP) and which set the fastest (Figure 5B). The mole% of PSer and the relative mole% of CS1 played a role in silicon release. With increasing the mole% of PSer and the relative mole% of CS1, the silicon concentration increased from the material (Figure 5C). 

#### 3.3.2. Effect of Volume/Surface Area on Ion Release 

One goal of this study was to identify the experimental conditions where cells would survive when in direct contact with PMCs using identical conditions in vitro as in vivo (e.g., no rinsing of samples). Therefore, the effects of sample dimensions and surface area to volume ratios on ion release were investigated next. Samples were fabricated according to ISO recommendations (8.9 mm × 4.5 mm “discs”, Figure 6), or the smallest dimensions that could be made using molds (1.6 mm × 1.6 mm, 4 mm^3^ volume, “cylinders”). As pH is a primary determinant of solubility (an uncontrolled variable in Figure 4), and solubility will affect the formation of precipitants, formulations #29 and #35 were selected for comparing the effects of sample dimensions on two different compositions that had a similar extract pH, composition (#29 has 6 mole% more phosphoserine than #35 (35.1 mole% vs. 29.9 mole%); for ICP analysis silicon was ignored as #35 did not contain silicon), and setting times. 

Generally, 10–12-fold more calcium was found in the soluble extracts from discs via ICP-OES, and 17–20-fold more phosphorus, compared to cylinders, during the first hour of release. Discs had a 4-fold greater surface area to volume ratios, and 16-fold greater mass per extract volume compared to the cylinders. After normalizing, we have results like for sample mass and extract volume (Figure 6A,B), and the discs released more phosphorus in soluble extracts for both formulations, at all time points, compared to cylinders. In contrast, the calcium concentration was identical in the extracts from the disc and cylinder for #35, while up to 30% less (cumulative) calcium was found in the extracts of #29 discs compared to cylinders after 72 h. The accumulation of calcium in the extracts from cylinders was higher than expected, compared to phosphorus. For both calcium and phosphorus, the formulation with a longer setting time and higher amount of phosphoserine, released more ions (#29 > #35). In short, the soluble fraction of ions depends significantly on the sample dimensions and volume, and this profile differs for each ion. 

#### 3.3.3. Effect of Volume, Cure Time, and Dilution Volume 

Next, the effect of cure time on the rate and amount of ion release was compared (Figure 7, cumulative release, Appendix A, non-cumulative release) in cylinders. In this study, PMCs were modeled for use as an “in situ” setting material, where curing occurs contiguously with ion release after implantation. In the clinic, and in vitro, PMCs can also be used as a “prefabricated” fully cured material [4], which we modeled in Figure 5, using cylinders that were cured for 24 h (“prefabricated”) or cured to the initial setting time (“in situ”). The release rate of ions was compared between “in situ” and “prefabricated” cylinders, and between formulations #29, #35, and a formulation containing solely CS1 as a calcium source (#34), to identify the source of phosphorous ions (e.g., without calcium phosphate, all phosphorous ions released from formulation #34 must originate from diffusion of phosphoserine). As the majority of ion release occurred within the first 24 h and prior mechanical testing suggested maximum setting and compressive/adhesive strength is attained within 24 h, for evaluating the effects of curing time on ion release, analysis was restricted to 3 days. It should also be noted that Figure 7A2–C2 directly corresponds with the exact conditions tested in Figure 8 (e.g., same formulations, same sample setting time, handling, and dimensions), therefore the ion release profiles shown in 30–80% less calcium (Figure 7A1 versus Figure 7A2), 30% less phosphorus (Figure 7B1 versus Figure 7B2), and 10–30% less silicon (Figure 7C1 versus Figure 7C2) were present in the soluble extracts from prefabricated samples, cumulatively after 3 days, compared to the in situ samples. Figure 7A2–C2 should reflect actual conditions during the first 3 days of our ALP studies. 

These results are in agreement with previous studies in calcium phosphates, which reported that ion release decreases proportionally with the set time [2]. Within the first hour, 50 ppm (prefabricated)–100 ppm (in situ) of calcium and phosphorus were released, dropping to 3–5 ppm by 24 h and 1–3 ppm by 72 h. Surprisingly, even though #35 contained only calcium silicate and no calcium phosphate (all phosphorus elution must arise from phosphoserine elution), comparable concentrations of phosphorus were found in the soluble extracts of both #34 and #35 (in situ), while #35 (prefabricated) released more phosphorus than #34 after 1–3 days. This result provides strong evidence that a significant portion, perhaps the majority, of phosphorus release arises from phosphoserine elution. Cumulatively for #34 (in situ), 8 mM of phosphorus was present in the 2 mL extract volume, which is equivalent to 16 micromoles of phosphoserine, out of a starting amount of 162 micromoles of phosphoserine in the PMC cylinder, representing 9.8% release of the total amino acid content over 3 days for formulation #34. This does not take into account the amount of ion/amino acid sequestered in any precipitants (insoluble fraction). In effect, PMCs adhesives are effective, chronic “drug delivery” devices for ions as well as amino acids (phosphoserine). 

When comparing the amount of silicon to calcium released, roughly 7-fold more calcium ions were detected per molecule of silicon, in both in situ and pre-fabricated samples (Figure 7A1 vs. Figure 7A2, and Figure 7C1 vs. Figure 7C2). There are only two precursors in #34, phosphoserine and calcium metasilicate. The calcium and silicate are originally present in a 1:1 molar stoichiometry, and for each molecule of silicon that elutes, an equivalent molecule of calcium must be released via dissolution. A comparison of Figure 7A1,A2 suggests that either more calcium is released, or diffuses more easily, relative to silicon, or that more silicon is precipitated or sequestered from the extract media. Despite the many limitations of this test model (e.g., not perfect sink conditions), the release curve of silicon appears almost linear, similar to a zero-order release (concentration independent). This suggests that silicon release may involve a non-diffusion related mechanism, like dissolution dependency, as no other silicate phases are expected to precipitate directly from pseudowollastonite. Formulation #34 allows us to track phosphoserine release kinetics, which appear fickian (diffusion-based, similar to time dependent kinetics, e.g., higuchi model), and compare them with phosphorus release from formulations containing both phosphoserine and calcium phosphate (#35). In contrast to calcium, where initial set time cured samples exhibited significant differences between groups (Figure 7A1) while 24 h cured cylinders released identical calcium concentrations (Figure 7A2), the 24 h cured #35 released more phosphorus than #34 and initial set time cylinders released identical amounts of phosphorus for #34 and #35. #35 contained 30 mole% phosphoserine, #34 contained 25 mole% phosphoserine, and #35 released approximately 50% more phosphorus than #34, cumulatively, from prefabricated cylinders, after 3 days. 

Most importantly, the pattern of ion release (#34 > #29 > #35 for calcium, #29 > #35 > #34 for phosphorus, and #34 >> #29 > #35 for silicon) was identical when comparing the in situ and prefabricated samples. This result suggests that the relative trends of ion release appear to remain the same for samples that are cured for short and long times (initial setting vs. completely cured), even in solutions where ion concentrations exceed solubility (precipitation was observed forming in all samples), for up to 3 days. These results provide preliminary evidence that may also predict relative trends of ion release, between different formulations, in vivo. For silicon release, formulation #34 contained significantly more silicon, and unsurprisingly released more silicon than #29 or #35. 

The amount of ions present in the extract media also differed between silicon and other ions; the elution rate of silicon was significantly greater at later time points (60–70% of cumulative release occurred during the later 1–3 days) compared to the release of calcium or phosphorus during this time period, for both in situ and prefabricated samples. These results may, in part, reflect the experimental conditions—the release rate is limited by the buffering capacity and precipitation rate of the medium (DMEM). In conditions where the media are replenished more frequently, the difference in ion release, between in-situ cured and prefabricated (24 h) samples, is likely to be larger than reported here, although the relative trend between formulations is expected to be similar. 

In summary, the ion release trends of in situ samples were similar to prefabricated samples, with 30–70% difference in cumulative (soluble fraction) ion release between in situ curing and fully cured samples. Sustained release of phosphoserine was verified in samples lacking calcium phosphate (#35), and the release profile of silicon differed from calcium and phosphorus, with a faster release at later time points. 

### 3.4. Cell Viability 

#### 3.4.1. MDPC23 Viability Screening (Indirect Contact) 

Three cell lines were selected to represent tissues that are most likely to come into contact with PMCs: MDPC23 odontoblasts (dental), MC3T3 pre-osteoblasts (bone), and L929 fibroblasts (connective tissue). The MDPC23 cell line was used to screen most formulations (Table 1), using undiluted, neutralized extracts (indirect contact) (Figure 8A shows the 1-day viability with corresponding compositions), before narrowing the number of formulations for testing in MC3T3 and L929 cells. The ISO standards for toxicity set 70% cell survival as the limit for “non” cytotoxicity. The viability data shown in Figure 8B are arranged by formulation number (Table 1). All groups, except #36 and #38, were cytocompatible after 24 h exposure to undiluted extracts. After 48 h exposure the average survival was greater than 70% for groups #42, #29, #35, #28, #27, #30, #31, and #39. Many of the formulations appeared to stimulate proliferation slightly after 24 h, though the difference was statistically significant only when comparing #31 to control (ANOVA, Dunnets Test post hoc, *p* < 0.05). None of the extract groups stimulated proliferation after 48 h. 

#### 3.4.2. MDPC23, MC3T3 and L929 Viability (Indirect Contact) 

Four formulations were selected for cytotoxicity testing in three cell lines (Figure 8C): #29, #34, #35, and #38. Formulations #29, #34, and #35 were either: tested previously in vivo or in vitro and were histo- and cyto-compatible, or used only calcium silicate (#34) as the calcium salt. Formulation #38 was included as an internal “positive control” as it appeared cytotoxic in the MDPC23 cells. There were similar trends between cell lines, with formulation #38 causing significant cytotoxicity in all cell lines. Interestingly, the MC3T3 cell line appeared the most sensitive, with the lowest survival after exposure to #29 and #35 (40–50% survival at 48 h), compared to L929 or MDPC23. In contrast, all three formulations were non-cytotoxic after 24 and 48 h exposure, in L929. After 48 h exposure, group #34 (silicate) appeared slightly toxic in MDPC23, but was the most cytocompatible formulation in all cell lines and time points. In summary, formulations where the calcium salt is completely replaced with calcium silicate (#34) were tolerated, as well as calcium phosphate PMCs, in vitro. Formulations that are known to be histo- and cyto-compatible in vitro and in vivo may appear cytotoxic (60–70% survival) when extracts are collected before the adhesive is completely cured. Considering the large amount of ions that are released during curing, these cytotoxicity results suggest that most formulations are likely to be well tolerated in vivo, even during the in situ curing phase. 

**Figure 8 biomedicines-10-00736-f008:**
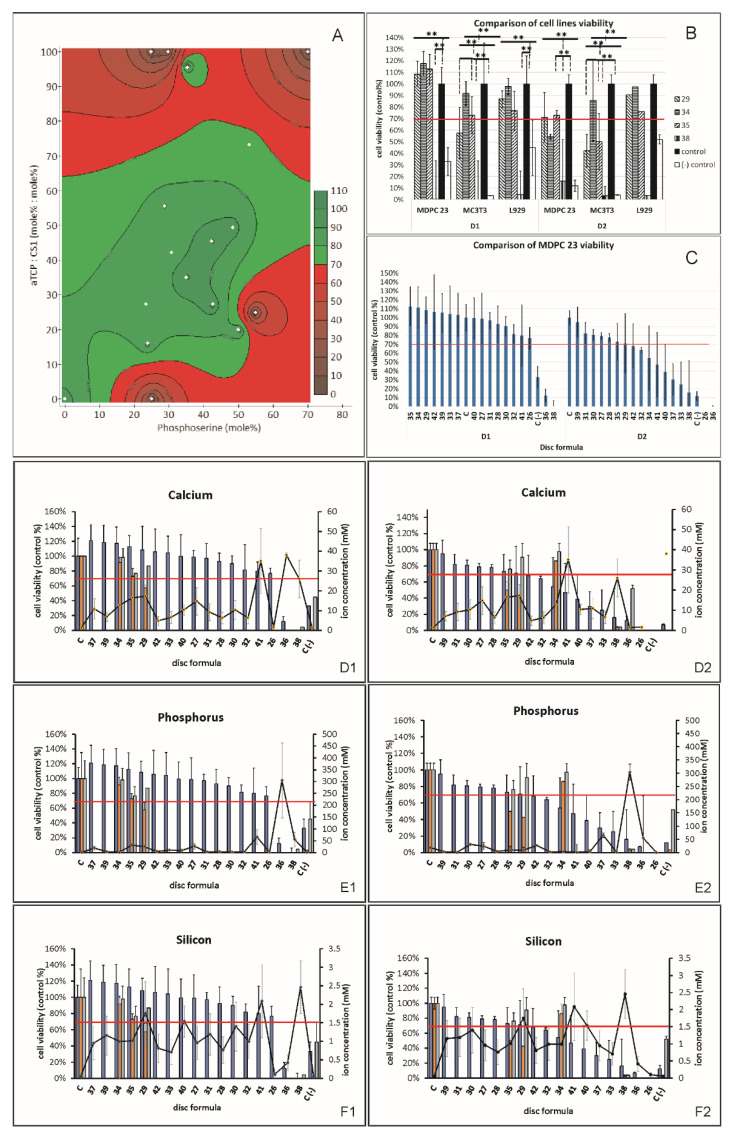
Cell viability in indirect contact with PMC extracts. (**A**) Formulation plot comparing MDPC23 viability (Z-axis) after exposure to extracts for 24 h, with adhesive composition (X- and Y-axis). Regions in green indicate conditions that are cytocompatible (cell viability ≥ 70%), while red regions represent opposing extremes that are cytotoxic (cell viability ˂ 70%). (**B**) Viability of MDPC23 after 24 or 48 h of incubation with extracts, the red line indicates cytotoxicity according to ISO-10993 (<70%). (**C**) After narrowing formulation range, comparison of 24 and 48 h viability, between MDPC23, MC3T3-E1, and L929. (**D1**–**F1**) Overlay of viability, after 24 or 48 (**D2**–**F2**) hours of exposure to extracts, with a concentration of leached (**D**) calcium, (**E**) phosphorus, or (**F**) silicon from ICP analysis. Sample dimensions are 8.9 mm diameter × 4.5 mm thick, cured to an initial setting time. Negative control is 10% DMSO. ** indicate *p* < 0.01, for comparisons made between: all groups compared to the control (ANOVA, Dunnets post hoc) in (**B**,**C**); or between all groups and the most toxic formulation (#38) in (**C**) (Games−Howell).

#### 3.4.3. Correlation/Regression Model 

In an attempt to correlate cell survival with each potential covariate, the following covariates were assembled into a statistical regression model: amount of Pser, CS1, and αTPC; pH; setting time; ion concentration for Ca, P, and Si; and output measure viability. Figure 8C–F compares the ion concentrations with the viability, for each formulation, in all cell lines. The correlation between silicon and viability appeared weak, as formulations eluting both high and low amounts of silicon (#36 and #38) exhibited a low viability (after 1 day). For both calcium and phosphorus, the ion concentration appeared to correlate with toxicity, as the formulations with the highest release rate (#36 and #38) appeared the least viable after 1 day, with the exception of formulation #41. Despite having comparable or greater calcium and phosphate release, formulation #41 was less cytotoxic than #38, although #41 also eluted roughly four-fold higher levels of silicon (could this be protective). 

### 3.5. ALP 

MDPC23 cells were incubated in direct contact with cylinders (Figure 6A) that had been cured for 24 h, and the protein levels (BCA) and alkaline phosphatase activity (ALP). Preliminary experiments confirmed that direct contact was cytotoxic if the smallest samples (cylinder) were cured to the initial setting time before contact with cells. Therefore, prior to ALP studies, the ion release profiles were compared between (a) in situ and prefabricated samples (Figure 6, how curing time affects ion release), and (b) between the cylinder samples used for direct contact (Figure 9) and the larger disc samples used for cytotoxicity (Figure 8) (how sample size and SVR ratio affect ion release) to allow for a later comparison between viability results (discs, in situ), ALP results (cylinders, prefabricated), and ion release studies. Visually, no toxicity or disruption of the cell sheet that forms during differentiation, or excessive mineralization occurred near the cylinder (Figure 8C and Figure 9B). This result suggests that any ALP expression changes do not reflect the localized phenomenon that only occurred in regions where the PMCs cylinders physically contacted cells, or from degradation or large ion release leading to isolated regions of mineral formation. 

In MDPC23 odontoblast cells formulation #34 (only silicate) and #35 (no silicate) consistently produced higher peak ALP, compared to control differentiation media (day 7 ALP in OM1, and day 3 ALP in OM2 conditions). #34 stimulated 65% more ALP activity than the control on day 7 (OM1), and the difference was statistically significant (ANOVA, *p* = 0.003, Dunnets post hoc, #34 vs. control, *p* = 0.012). Interestingly, while the higher ALP activity might be explained by the greater silicon release in #34; formulations #35 and #29 had identical ion release profiles to each other for calcium, phosphorus, and silicon (Figure 6), with #29 actually releasing about 10% more of each ion than #35, while #29 stimulated a lower ALP activity than #35. Rather than comparing ion release, if one compares the starting amount of phosphoserine in each cylinder with the ALP activity (#34 (25 mole%) < #35 (30 mole%) < #29 (35 mole%)), it matches closely (inversely proportional) with the trend in ALP activity on day 7 (OM1) in both MDPC23 and MC3T3 cells. While #35 and #29 had identical ion release profiles, it is possible that a greater percentage of phosphorus was represented by phosphoserine release. These results suggest that there may be a negative correlation between phosphoserine release from PMCs and cell differentiation. 

Separately from the values shown in Figure 9C, which are ALP activity normalized to protein concentrations (BCA assay), the protein levels were also compared to identify any difference between groups (Figure 9B). All PMCs treated MDPC23 groups produced 16% less protein (1100 ug vs. 1300 ug, Dunnet’s post hoc, *p* = 0.004 #29, 0.027 #34, 0.028 #35) on day 3, and 27% and 17% less protein on day 7 (*p* = 0.001, 0.005 for #29, #34 vs. control). In MC3T3 cells, only group #29 treated cells exhibited a 23% lower protein content compared to the control at day 7 (Dunnets post hoc, *p* = 0.014). Lower protein levels did not correspond with higher ALP/BCA results. 

## 4. Discussion

There are a number of unmet challenges facing the fields of regenerative medicine and biomaterials. One of the challenges is the poor predictive power of in vitro testing, which necessitates extensive testing in large animals before moving to human testing [36], and the prohibitive cost of evaluating more than one or two device iterations in vivo. For every device that reaches the final stages of in vivo testing, hundreds of potentially better performing variants are neglected (untested) because of cost, time, and ethical concerns (e.g., hundreds of animals cannot be used to evaluate all iterations of a device). This is a concern for PMCs because of the large number of unique formulations that are possible by changing the ratios of phosphoserine and calcium salts. Each formulation will have a unique ion elution profile and, potentially, beneficial cellular effects. However, it is impossible to evaluate safety and efficacy for more than a handful of formulations, when a typical animal study might only allow for 2–3 implants per animal. In the present study, we evaluated many PMCs formulations, ex vivo and in vitro, to identify compositions with interesting properties. 

While in vitro testing with cell cultures gives fast and very specific information on distinct, cell-level processes, in vitro assays are often not predictive of how safe or effective a device will be in vivo [36]. Therefore, while biochemical screening tests, at the cellular and molecular level, should be used prior to clinical testing in order to screen for problems with the biocompatibility of implanted devices [48], in vivo testing is necessary [12]. Of the ions PMCs release, calcium has the strongest correlation with new bone formation in vivo and is one of the most clearly understood mechanisms underlying direct effects (Table 4), via the calcium sensing receptor (CSR). These studies in Table 4 show that increases in bone formation in vivo are correlated with chronic calcium release in the range of 0.5–20 mM [49,50,51,52,53,54]. Excessive calcium is cytotoxic in vitro above 10 mM [55], but little information is available on the local cytotoxicity, or inhibitory dosage, in vivo. 

In the present study, prefabricated PMCs released up to 2 mM (0.26 mmol·g^−1^, 1.6 mmol·mm^3^) of calcium during the first hour, 10 mM within 24 h, and 1 mM chronically every 24 h. The calcium release rate was in the range needed to stimulate the chemotaxis of monocyte/macrophages, MSC, and osteoblasts in vivo period. Calcium concentration also stimulates angiogenesis behavior and osteogenesis via differentiation/polarization of monocyte/macrophages, MSC, osteoblast, endothelial, and osteoclast cells. Calcium release from PMCs can be finely tuned by altering the composition (e.g., add more calcium salt, or a more basic calcium salt to alter the pH) and setting conditions. When comparing the chronic release rate of calcium between PMCs and other biomaterials, prefabricated ion-coated polymers release 0.2 mM [56,61] to 2 mM [50], bioglass (BG45S5) releases 33 mM [62], sintered hydroxyapatite discs and scaffolds release 0.05–0.25 mM [51,52,54,57,63,64], and tricalcium phosphates release 0.1–2.5 mM of calcium each day. In short, fluids incubated with fully cured PMCs contained 1–20 fold more (soluble) calcium than “osteoinductive calcium” phosphate bioceramics, up to half as much as osteoinductive polymer composites, and 30-fold less calcium than bioglass when the media were replenished frequently. 

Out of the numerous in vitro studies reporting ion release profiles for materials like bioglasses, ion coated polymers like poly lactic acid (PLA), and cements, it is difficult to compare those results with our results, because each study uses slightly different sample preparation and ion release methods, or reports the results incompletely (e.g., mg/kg release units do not account for media volumes; molarity (mM) does not account for surface area, porosity, etc.). Most importantly, when comparing ion release between prefabricated and in situ curing samples, in situ PMCs released up to 2–3-fold more calcium during the first hour (4 mM) and 20% more calcium chronically (2 mM per day), than prefabricated PMCs. Since we observed a large amount of ions sequestered as precipitants, which were not quantified in this study, the true amount of calcium released from PMCs in vivo is likely to be much higher than indicated in this study (likely 2–10-fold higher based on the relative amount of precipitants we observed). It is possible that a similar precipitation phenomenon will occur in vivo at the site of implantation, possibly facilitating the formation of a mineralized hydroxyapatite layer between implanted PMCs and bone surfaces [30]. 

Silicon and phosphorus are also strong stimulators of bone formation. This is known regarding the biological mechanisms, and the effects are often indirect as there is no analogous calcium sensing receptor signal transduction pathway for silicon and phosphorus. Implanted phosphorus (2.5 mM) [65] or silicon ions (6 mM) [66] can stimulate mineralization in vivo. Silicon ions stimulate mineralization in vitro via direct osteogenic gene expression (e.g., alkaline phosphatase increase, in the range of 0.02–2.5 mM silicon in osteoblasts and MSCs [53,67,68,69], and via cell differentiation of fibroblasts, endothelial, and dental pulp cells towards angiogenesis behavior and blood vessel (tubule) formation (0.1–1 mM)) [53,70,71]. The authors were unable to find any studies describing the effect of silicon ions on inflammatory or bone resorbing cell types, or chemotaxis/cell recruitment in response to silicon gradients. Phosphorus ions stimulate the recruitment/chemotaxis (4–10 mM) [72] and osteogenic differentiation of osteoblasts and MSCs (0.5–10 mM) [72,73,74]. While it is clear that phosphate eluting ceramics can directly affect inflammation and angiogenesis [57], the authors were unable to find any studies describing the direct effects of phosphate ions on the recruitment or differentiation of inflammatory, angiogenic, or resorbing cell types. 

In this study, prefabricated PMCs released up to 0.1 mM of silicon after 1 h, 1 mM after 24 h, and 0.5 mM chronically every 24 h. In comparison with other biomaterials, silicon containing polymer/hydrogels release up to 0.5–1 mM [75,76], prefabricated bioglasses release up to 1.4 mM [62,77], and biphasic calcium phosphates release 5.7 mM [53,66] of silicon each day. While prefabricated PMCs released less silicon that other biomaterials, below the range reported to stimulate mineralization in vivo (6 mM) [66], in situ curing PMCs released 50% more silicon, and silicon release from PMCs (#34) was sufficient to stimulate a higher ALP expression in odontoblasts in this study. 

Prefabricated PMCs released up to 0.3 mM of phosphorus after 1 h, 5 mM after 24 h, and 1 mM chronically every 24 h. In comparison, phosphorus containing polymer/hydrogels released up to 0.2–1 mM [61,74], prefabricated bioglasses released up to 1 mM [62,78], and biphasic calcium phosphates released 0.01–0.3 mM [57,64] of phosphorus each day. Prefabricated PMCs released as much, or up to 5-fold more phosphorus that other biomaterials, and in situ curing PMCs released up to 25% more phosphorus than prefabricated. The concentration of soluble phosphorus released by PMCs (1–8 mM) is sufficient to stimulate mineralization in vivo, and osteogenic cell recruitment and differentiation in vitro. 

Ion release studies also typically require extraction media compositions, extract volumes, sample preparation, and rapid replenishment of media, such that ion concentrations remain far below solubility (perfect sink conditions), where precipitants will not form. The media pH are also controlled in release studies, via strong buffers that are often cytotoxic (e.g., Tris). While in vitro studies of ion release, bioactivity, and cytotoxicity also use extraction media and sample preparation such that ion concentrations remain far below solubility, media are replenished slowly in static cultures, leading to the rapid accumulation of ions and precipitants. In vivo, PMCs are intended for use in confined spaces, where reconstructed tissue surfaces may receive limited fluid flow until new vasculature develops. Similar to the present study, fluid replenishment at injury sites in vivo will occur slowly, involve supersaturated fluids, and eluted ion concentrations during the in situ curing period (first 1–24 h) are likely to exceed solubility, leading to either precipitation or bioactive mineralization. 

We observed a 4-fold increase in ions when samples were cured in situ vs. prefabricated, in the first hour of release, 2-fold in the first 24 h, and 10% chronically every 24 h over 7 days. This suggests that in vitro testing is likely to underestimate in vivo ion release and biological effects. Over the range of 10 mM to 12 mM calcium release, cells cultured on prefabricated PMCs produced more ALP. This suggests that actual in vivo in situ curing samples may release too many ions. Therefore, in vivo testing should include formulations that specifically release 2–10-fold differences in ion release to identify which formulations and ion concentrations are beneficial in vivo, rather than in vitro. Therefore, we conclude that the observed concentrations of ions in this study were more likely to reflect physiological events (in vivo) than if we had used routine in vitro (pre-fabricated and pre-leached discs, ISO-10993) or routine release study methods (e.g., perfect sink conditions). 

In addition to being the first study to report beneficial ion release from phosphoserine tissue adhesives, this is the only study to evaluate ion release during the in-situ curing phase of an adhesive biomaterial. In this study, we identified how differences between in vitro and in vivo-like test conditions affect ion release from PMCs. We observed higher soluble calcium release for larger surface area samples, while more soluble phosphorus was detected with the increasing sample mass, rather than surface area. As PMCs are used as a thin layer adhesive (<1 mm thick), we would expect a higher calcium release, and less phosphorus per gram of adhesive in vivo, compared to in vitro testing. Likewise, in vivo, based on the results of the present study, we would expect thin layers of PMCs to release 50–200% high concentrations of calcium, phosphorus, and silicon during the first 24 h of in situ curing, compared to in vitro studies that use “prefabricated” samples. We have previously reported that PMCs degrade slowly ex vivo (10% mass loss over 4 weeks [5], with much faster degradation when implanted in bone in vivo (50–75% volume changed to bone within 12 weeks in rodents [24], or within 52 weeks in rabbits [13]) and negligable degradation in vivo when PMCs are implanted in tissues that lack mineral resorbing cells [4]. Taking into account cell mediated resorption, the ion release profiles are expected to be even higher in vivo, chronically, than we can detect using in vitro methods. 

We have also shown that PMCs can stimulate osteogenic protein activity (ALP) and that the amount of phosphoserine and ions that are released are sufficient to explain this phenomenon. Alkaline phosphatase is an early differentiation marker in mineralizing cells, which catalyzes the conversion of organic phosphate sources into inorganic orthophosphate ions, thereby creating a supersaturated environment that facilitates spontaneous precipitation of a new mineral [38,79]. When added to the extracellular medium, phosphoserine accelerates mineralization in mouse osteoblast (MC3T3-E1 [19] and human stem cells (adipose derived stem cells, hADSC [20]) at concentrations of 1.7–3.5 mM in vitro. In vivo, 1.7–3.5 mM phosphoserine also enhanced new bone formation more than 2-fold [19]. In this study, 1–5 mM of phosphoserine, released from #34, stimulated ALP in odontoblasts but not in osteoblasts, in contrast with the results of Park et al. [19]. This could be because we evaluated only early protein activity (up to 7 days), while Park evaluated only late expression and mineralization (14–32 days). In MC3T3 cells, peak ALP expression typically occurs in the first 7–10 days [80], hence we focused on 3–7 day expression. The effects of the phosphorus source on differentiation and mineralization must also be considered [81]. Park et al. [19] has shown that osteogenic gene expression is comparable in MC3T3 when either beta glycerol phosphate or phosphoserine are used as organic phosphate sources. In this study, when phosphorus release from PMCs was solely phosphoserine (#34 did not contain calcium phosphate), ALP activity was stimulated as strongly as for PMCs releasing only inorganic phosphorus (#35). It is therefore possible that even though #29 and #35 had similar ion and phosphorus release profiles, the actual amount of phosphoserine was significantly different, which could explain why #35 stimulated higher ALP in MDPC23 and #29 did not. It should be noted that the ALP activity appeared to be inversely correlated with the phosphoserine content; lower phosphoserine containing PMCs stimulated higher ALP in odontoblasts. 

Because of the high concentration of ions released in this study, we did not conduct mineralization studies (e.g., alizarin red or von Kossa staining), as it would be impossible to differentiate between contributions from spontaneous precipitation (supersaturation) on secreted collagen matrix proteins, nucleation from apoptotic cells, or from cell differentiation and enzyme activity (ALP). We also did not observe an increase in proliferation in any of the three cell types, over a wide range of PMC formulations. Ying et al. [20] also reported no change in stem cell proliferation at concentrations of phosphoserine similar to our study. However, the authors of [31,76] reported increases in osteoblast and odontoblast cell proliferation in response to calcium, phosphorus, and silicon ions in the same range as our studies [31,41,69]. 

Phosphoserine modified cements have been reported to improve healing and mineralization in vivo as early as 2005 [17,18]. However, all studies prior to 2018 only used phosphoserine concentrations below 10% (weight%), where phosphoserine did not create tissue adhesive effects, and the cement physical properties (e.g., density, morphology, microstructure, strength) are significantly different from modern PMCs [5,13,14]. Our PMCs use 2–5-fold more phosphoserine (weight%) and likely release more bioactive ions and soluble phosphoserine. Therefore, the in vivo, biological responses to modern, higher phosphoserine containing PMCs may differ from prior studies. 

There were many limitations in the present study: the foremost being the use of supersaturated conditions during ion release, the formation of precipitants, and the lack of quantitative data on how much ion(s) were unaccounted for in the insoluble precipitant fraction. Since the precipitants were not collected and analyzed by ICP, the actual ion release was much higher than we report here. Furthermore, since the release studies were not conducted under “ideal sink” conditions, the “true” diffusion-based rate of ion release, from PMCs, is likely much higher than reported here, with potentially larger differences between different formulations than observed in this study. The precipitants were not measured, as removing them during the release study (at each time point) would affect the release rate (e.g., shifting the supersaturation and precipitation equilibrium via Le Chaterle’s principle), and waiting to quantitate the precipitants after the last time point would not give information about ion release for each earlier time points. The purpose of this study was to collect preliminary data, where no other data exist, on ion release and cell response under conditions that mimic the actual in vivo condition. An in vivo study has recently been completed [24] in rodents, and one is ongoing in sheep, evaluating in-situ curing PMCs with identical dimensions as in this study (approximately 1–2 mm diameter). The fluid exchange and sample preparation conditions for these in vivo studies are expected to closely mirror the test conditions of this study. Therefore, we hypothesize that similar precipitants will occur in vivo, where blood and fluids are supersaturated, and that these may contribute to tissue integration and remodeling. Lastly, the pH was not controlled in our ion release studies, which will affect the solubility of the precipitating phases, as well as ionic states of ionic species (e.g., H_2_PO_4_^−^ vs. H_3_PO_4_) [32,82,83,84,85]. The pH is determined by chemical composition of PMCs, particle size, accelerators, additives and retardants in the liquid phase, liquid type, temperature, etc. [3,10,86]. Considering the complexity of designing and testing different PMC formulations, we restricted our study to compositions that were cytocompatible, with an initial setting time close to 5 min and close to neutral pH. We evaluated PMCs containing silicates because the higher pH and faster dissolution of silicate salts can neutralize the highly acidic phosphoserine, and allow for finer control of handling/setting, and potentially create osteogenic properties [2,3,12,31,87,88,89,90].

## 5. Conclusions

We studied a class of bone tissue adhesives (PMCs). A wide range of formulations of PMCs are cytocompatible when in indirect contact with fibroblasts and in direct contact with osteoblasts and odontoblasts. Each unique adhesive formulation released calcium, phosphorus, phosphoserine, and silicon ions at different rates. Unique PMCs formulation released ions in higher concentrations that other ceramics and biomaterials that are considered osteoinductive or osteostimulatory. Finally, PMCs can stimulate higher alkaline phosphates activity in osteogenic cells, possibly contributing to osteointegration in vivo. Consequently, future studies on how PMCs affect cell and tissue scale responses, in vivo, are warranted. 

## Figures and Tables

**Figure 1 biomedicines-10-00736-f001:**
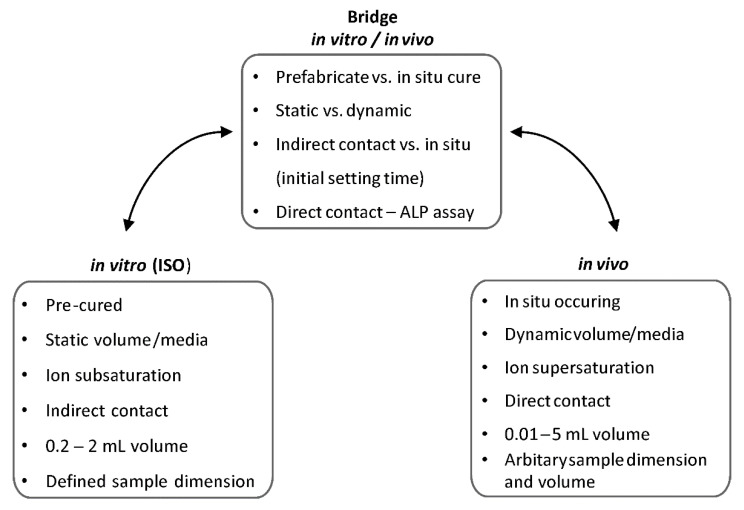
Comparison of common study designs and the present study.

**Figure 2 biomedicines-10-00736-f002:**
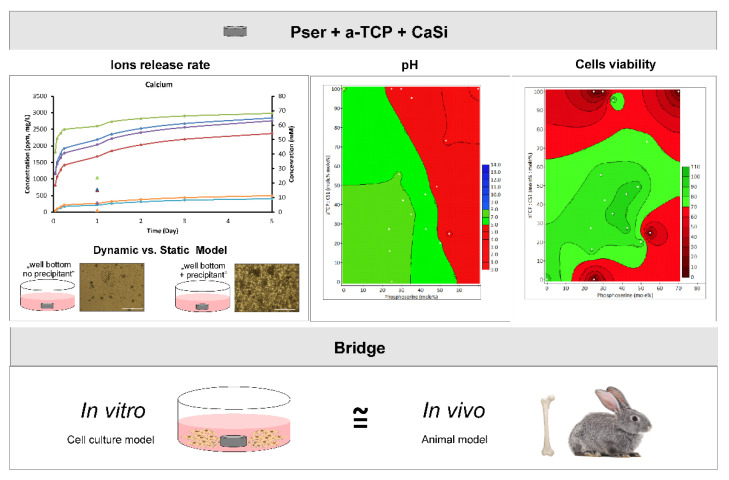
Scheme of the experimental design—simulation of conditions similar to in vivo.

**Figure 3 biomedicines-10-00736-f003:**
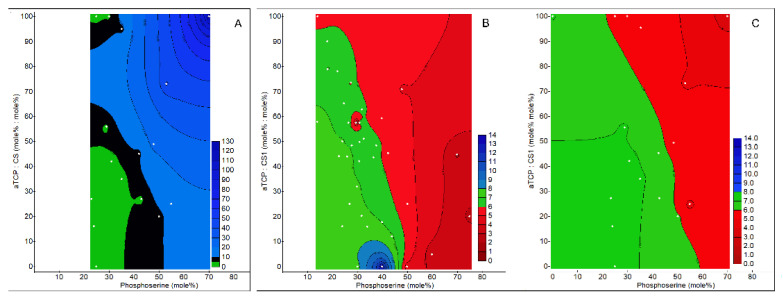
Formulation plots comparing the effect of composition on setting time and pH, in “as obtained” extracts, according to ISO 10993. (**A**) Initial setting time (Gilmore needle) for PMCs formulations (<2 min setting = green, 2–10 min = black, and >10 min = blue regions). pH of extracts when samples were cured (**B**) for an arbitrarily fixed time (5 min), or (**C**) to the initial setting time for each formulation (pH ˃7.6 = blue, pH < 7.2 = red, pH 7.2–7.6 = blue region). Sample dimensions were 8.9 mm × 4.5 mm (disc), incubated in 0.2 g mL^−1^ of media.

**Figure 4 biomedicines-10-00736-f004:**
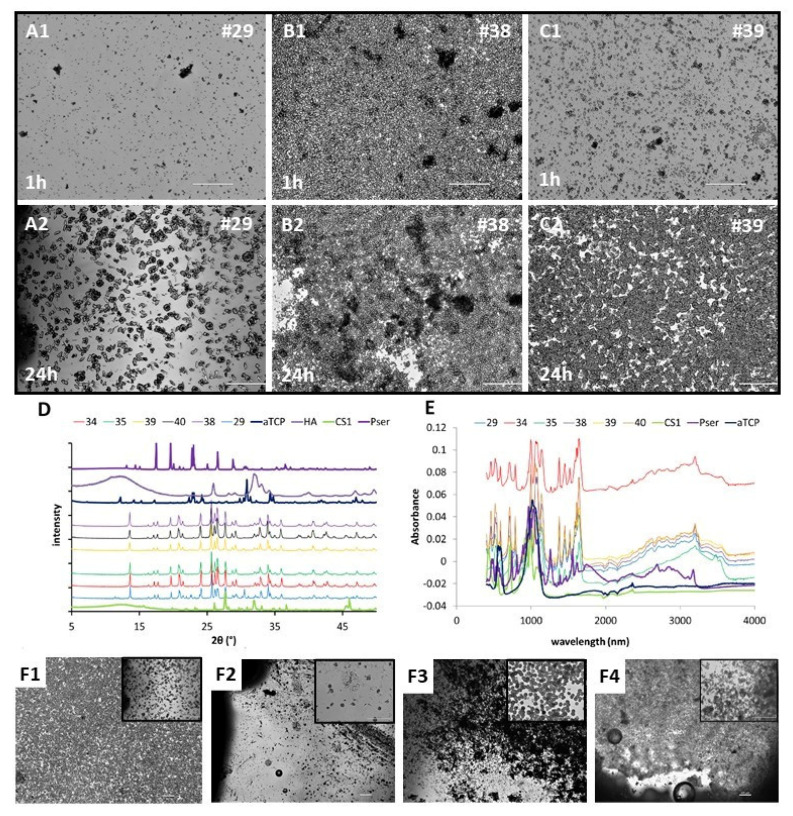
Light microscopy images and chemical analysis of precipitants. (**A1**–**C1,A2**–**C2**) Precipitant formation/density and effect of immersion time. (**D**) XRD and (**E**) FTIR spectra of dried precipitant powders, collected after sample immersion for 24 h. (**F**) Light microscopy images of precipitants for the four most interesting formulations, #29 (**F1**), #34 (**F2**), #35 (**F3**), and #38 (**F4**), immersed in the medium for 24 h. For all inserts, sample dimensions were 8.9 × 4.5 mm (discs), cured to an initial setting time. Scale bar size is 200 µm.

**Figure 5 biomedicines-10-00736-f005:**
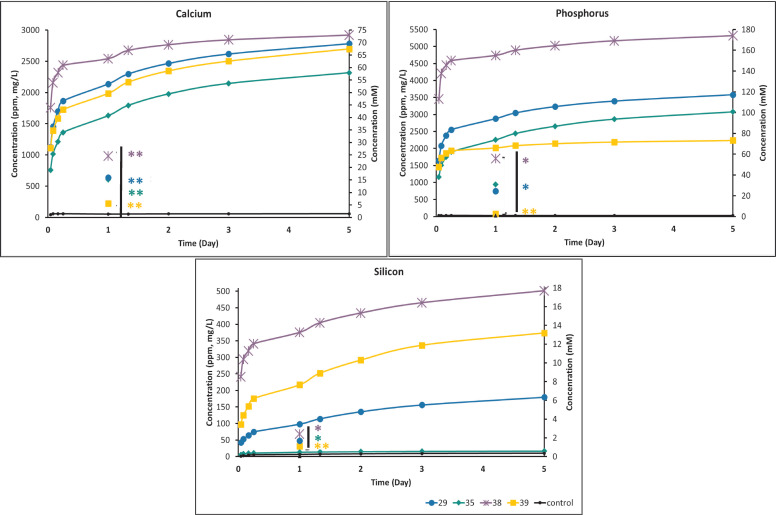
Effect of media replenishment on ion release profiles, for a broad formulation range. Ion release profile was measured in dynamic (lines) and static (single data points at 24 h) mode only for samples #29, #34, #38, and #39, and control for (**A**) calcium, (**B**) phosphorus, and (**C**) silicon. These samples were selected to represent the range of compositions and viability. Single data points at 24 h represent “static” data in accordance with ISO 10993 (media collected only once after 24 h immersion). Sample dimensions were 8.9 mm × 4.5 mm, cured to the initial setting time before immersion. **, and *, § indicates *p* < 0.01 and 0.05, respectively, for comparisons made between the cumulative release of each respective static or dynamic group at 1 day (*, Games−Howell); and between all groups and the highest eluting sample (#38, §, Dunnets post hoc).

**Figure 6 biomedicines-10-00736-f006:**
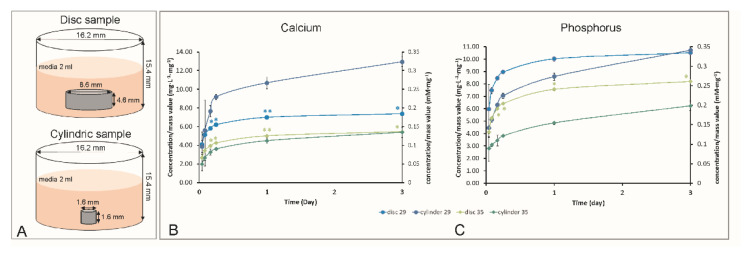
Effect of sample dimensions on ion release. (**A**) Graphical overview of sample dimensions and experimental comparisons. Comparison of ion release rates for (**B**) calcium or (**C**) phosphorus, between two PMCs formulations that previously showed significant differences in Figure 3 (#29 and #35), and between different sample dimensions (disc vs. cylinder) and surface area to volume ratios. Sample dimensions were 8.9 mm × 4.5 mm (disc) and 1.6 mm × 1.6 mm (cylinder), all cured to an initial setting time. The ion concentrations (Y axis) are normalized to sample mass (per milligram), as the final material densities differed between different formulations. ** and * indicate *p* < 0.01, and 0.05, respectively, for comparisons made between sample dimensions, at each time point (Games−Howell).

**Figure 7 biomedicines-10-00736-f007:**
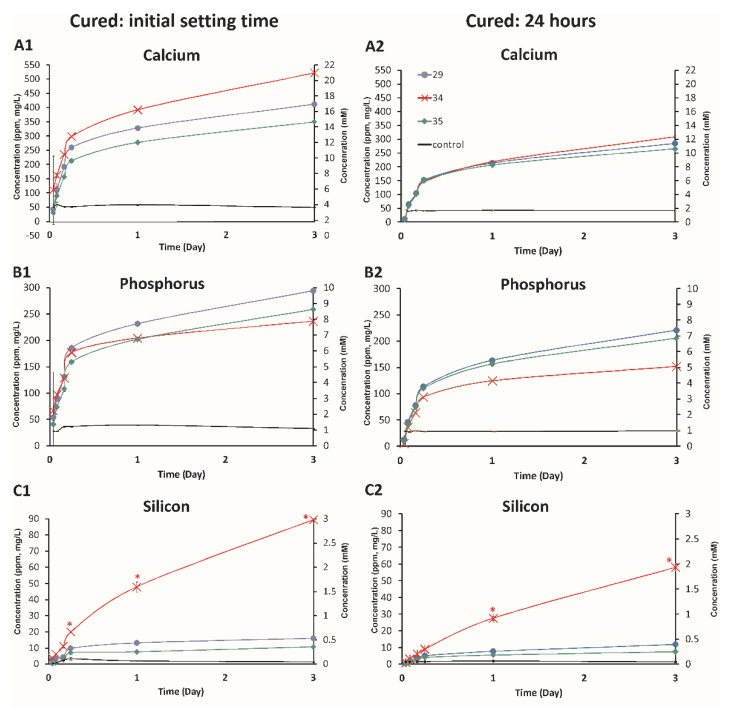
Effect of curing time on ion release profiles. Cumulative ion release profiles for calcium (**A**), phosphorus (**B**), and silicon (**C**) for samples cured for the initial setting time (**A1**–**C1**) or 24 h (**A2**–**C2**), via the ICP-OES analysis. Sample dimensions are 1.6 mm × 1.6 mm (disc). * indicate *p* < 0.01 for comparisons made between all groups, compared to the highest eluting formulation (#34) (Games−Howell).

**Figure 9 biomedicines-10-00736-f009:**
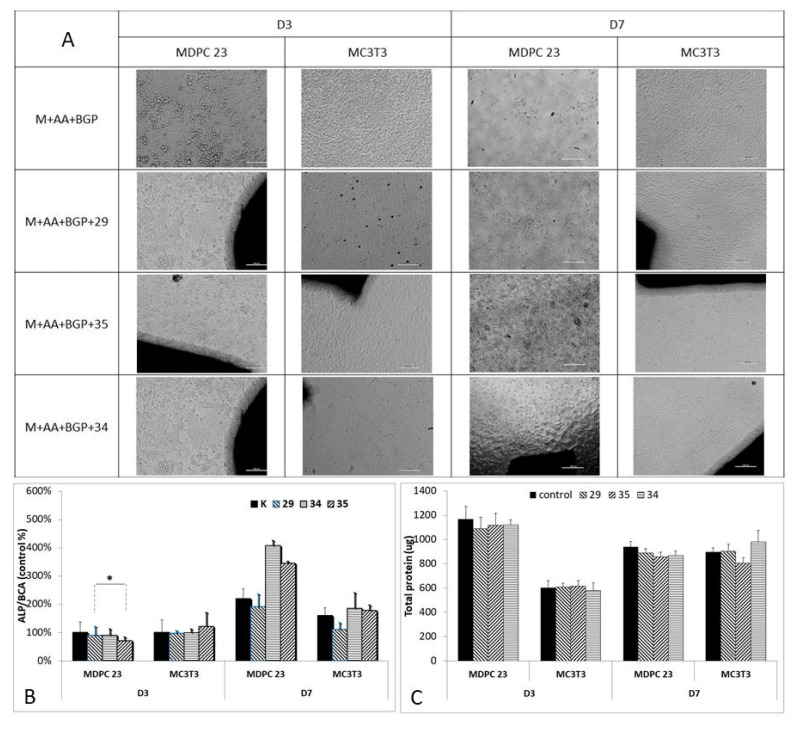
Morphology, alkaline phosphatase activity, and total protein content in MDPC23 and MC3T3-E1 cells after direct contact with PMC. (**A**) Light microscopy images of cells in direct contact with PMC, without visible cell death or excessive precipitation. (**B**) ALP/BCA values after 3 or 7 days. (**C**) Total protein content (BCA) of cells after 3 or 7 days. PMCs sample dimensions were 1.6 mm × 1.6 mm (cylinders), cured for 24 h before placing directly into wells with cells. M = medium, AA = ascorbic acid, BGP = β-glycerophosphate. * indicate *p* < 0.01, and 0.05, respectively, for comparisons made between formulation groups compared to the control, at each time point (Games−Howell).

**Table 1 biomedicines-10-00736-t001:** Table of disc formulations used for the Alamar Blue, ALP, and ICP experiments, including the setting time.

Sample	PSer	a-TCP	CS-1	pH	Cure Time (min)	Setting Time (min)		a-TCP Relative Moles%	CS-1 Relative Moles%
	Moles%	Moles%	Moles%			Inital	Final		
27	24.9	75.1		5.663	1.6	1.6	3.5	100	
28	28.8	39.7	31.6	4.409	1	1	2.4	55.7	44.3
29	35.1	61.6	3.4	5.771	4.4	4.4	9.6	94.8	5.2
30	23.3	20.9	55.8	7.936	1	1	1.4	27.3	72.7
31	24	12.1	63.9	7.974	1	1	1.6	15.9	84.1
32	30.7	29.2	40.1	7.609	1	1	2	42.1	57.9
33	42.4	26.1	31.5	6.270	5.6	5.6	15.6	45.3	54.7
34	25		75	7.527	1	1	1.6		100
35	29.9	70.1		5.552	2.4	2.4	7.2	100	
36	69.9	30.1		2.875	130	130	200	100	
37	48.3	25.4	26.3	5.495	26.8	26.8	120	49.1	50.9
38	54.9	11.2	33.9	3.988	16.8	16.8	46.5	24.8	75.2
39	35	22.7	42.3	7.375	1	1	2.8	34.9	65.1
40	50.1	10	39.9	6.188	6	6	24.5	20	80
41	53.3	34.2	12.5	4.523	59	59	122.5	73.3	26.7
42	42.8	15.6	41.6	6.9	1.8	1.8	9.2	27.3	72.7

Note that the final two columns, “relative moles%” refer to the X- and Y-axis of formulation plots in Figures 1A and 7, where the Y-axis reflects the molar ratio of aTCP to CS-1, and the X-axis reflects the molar ratio of phosphoserine to the total moles.

**Table 2 biomedicines-10-00736-t002:** Chronic ions release 1–5 days (mM).

	#38	#29	#35	#39	DMEM
Calcium	2.49–4.99				1.37
Phosphorus	1.61–6.45				0.96
Silicon	1.24	0.89	0.053–0.071	1.35	0.028

**Table 3 biomedicines-10-00736-t003:** Concentration of ion release at each time point after 24 h (mM).

	#38	#29	#35	#39
Calcium	1.87–2.49	3.74–4.49		
Phosphorus	4.84–6.45			1.61–2.58
Silicon	0.035–0.071	0.89–1.25	0.71–1.03	1.06–1.60

**Table 4 biomedicines-10-00736-t004:** Correlation of the calcium concentration and bone formation.

Function	Cell Type	Concentration	References
Calcium stimulates the recruitment and chemotaxis	monocyte/macrophages	1–8 mM in vitro and in vivo (equivalent to 0.1–1 mole of calcium released per gram of material)	[50]
	primary osteoblasts	2.5–5 mM in vitro and in vivo	[49]
	MSCs	3.5 mM in vitro, 10–20 mM in vivo	[49]
	osteoclasts or endothelial cells	no	
Calcium ions stimulate pro- osteogenic and angiogenic differentiation	monocyte/macrophages	6–8 mM acute, 0.25–2 mM chronic release, polarizes towards M2	[50,54,56,57]
	osteoblasts	0.2–8 mM stimulates ALP	[49,56,57]
	MSCs	0.5–2 mM 3–8 mM	[49,51,52,57,58]
	endothelial	4–6 mM HDEC in vitro and in vivo 0.017 mM in HAEC 10–20 mM in bone marrow progenitors	[50,53,59]
	osteoclast cells	2–6 mM	[60]

## Data Availability

All data generated or analyzed during this study are included in this published article.

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
