# Peer review of "Cytocompatibility and Bioactive Ion Release Profiles of Phosphoserine Bone Adhesive: Bridge from In Vitro to In Vivo"

_biomedicines, 2022, doi:10.3390/biomedicines10040736_

Round 1
Reviewer 1 Report
New bone replacement materials compatible with living cells are in great demand as scaffolding materials in regenerative medicine. The manuscript describes a thorough study on αTCP-based bone replacement material chemical properties. The study design is aimed on bridging in vivo and vitro testing. The manuscript is well-written and the results are presented in a clear way. The conclusions are suported by the experimental data. However, some improvements should be done before publication:
- The aim of the study should be reshaped to make the study more focused. In fact, the declared aims are interconnected and can be formulated a a single aim
- A large piece of literature data between the aim of the study and the final conclusion (ll 105-121) should be placed before the aim of the study
- The section describing statistical methods should be expanded. It is not clear from the text how many replicated were used in the experiments. E.g. it looks like only one sample of each composition was used in each experiment (though it is unlikely because 1way ANOVA was used). If so, some explanation should be given why the authors consider their results as representative. If more than one sample of a certain composition (e.g. 27, 28, 29 etc) was used in each experiment, then the number of replicates should be given in methods description and in figure legends.
- Figure 5 should be checked additionally. Were ion concentration measured in static and dynamic mode for each sample? Are all the lines presented in the chart or only 'static' (or dynamic) data? Are significance marks placed correctly (e.g., where are § signs shown in the legend?) ? The standard deviations should be shown as whiskers.
- There is no figure legend for Figure S3_2 (to judge from the file name). There is no figure for S2.2 figure legend. I suggest it is a typo in file name?
- Dental stem cells are known for their high osteogenic potential and high ALP level. Is it possible to add in Figure 9b the ALP level at the beginning of the experiment. The difference in Dex concentration for 3T3 and MDPC23 should be explained. Fibroblasts, MSC and dental stem cells are very sensitive to Dex concentration. High doses of Dex can induce adipogenic differentiation in some of cell lines of mesenchymal and epithelial-mesenchymal origin.
Author Response
Response to the reviewers’ comments
Dear Editor,
We have received your opinion and reviewers’ reports on our manuscript. We have appreciated the deep insight and effort of the reviewers and would like to thank them for all comments and recommendations which we have seriously considered in the revised version. A number of these comments were particularly relevant and helped to improve the original manuscript. We are very grateful to all reviewers for their revisions of our manuscripts. In what follows, we provide answers to the comments and a detailed explanation of how the reviewers’ recommendations have been reflected in the revised text. We have used the following formatting style to distinguish between the reviewers’ questions, our answers, and changes made in the main text: (1) Reviewer´s summarizing text is in italics, (2) questions and comments are in bold font while (3) our responses are in normal plain font and (4) all changes and modifications made in the text of the manuscript are yellow-colored.
Reviewer 1:
Comments and Suggestions for Authors
New bone replacement materials compatible with living cells are in great demand as scaffolding materials in regenerative medicine. The manuscript describes a thorough study on αTCP-based bone replacement material chemical properties. The study design is aimed on bridging in vivo and vitro testing. The manuscript is well-written and the results are presented in a clear way. The conclusions are suported by the experimental data. However, some improvements should be done before publication:
- The aim of the study should be reshaped to make the study more focused. In fact, the declared aims are interconnected and can be formulated a single aim.
Thank you for your recommendation. We edit the aims of the study. (introduction section, page 3)
The purpose of this study was to evaluate how different formulations of PMC affected: A) ion release profiles; B) cytotoxicity in multiple cell lines (MDPC 23, L929, MC3T3); and C) try to “bridge” in vitro and in vivo study methods. We have evaluated how different handling conditions, curing times, sample dimensions, formulations, etc. affected curing and ion release. We have also evaluated cytotoxicity to identify how formulation and composition affects the safety of PMCs (ISO 10993). Finally, we have tried to modify typical test methods (e.g. shifting from static to dynamic culture) to more closely represent in vivo conditions. In the present study, we use rapidly curing formulations in-situ, and attempt to overcome the limitations of static, mono-cell-type cultures, and to more accurately mimic the initial cell-material interactions that occur in vivo.
- A large piece of literature data between the aim of the study and the final conclusion (ll 105-121) should be placed before the aim of the study
Thank you for your recommendation. We edited the position of these parts in our manuscript. We placed part of the literature (original ll 105-121) before the aim of the study. (Introduction section, page 3)
- The section describing statistical methods should be expanded. It is not clear from the text how many replicated were used in the experiments. E.g. it looks like only one sample of each composition was used in each experiment (though it is unlikely because 1way ANOVA was used). If so, some explanation should be given why the authors consider their results as representative. If more than one sample of a certain composition (e.g. 27, 28, 29 etc) was used in each experiment, then the number of replicates should be given in methods description and in figure legends.
Thank you for your recommendation. We edit the part of the statistical analysis. (Material and Method section, page 6)
Statistically significant differences were identified with SPSS software (version 22), with 1-way ANOVA. Details are provided in the captions for each figure, and, briefly: ”In Figure 5, for each ion and at each time point, the cumulative ion concentration at 24 hours was compared between “static” and “dynamic” samples for the same group (e.g. #38 static vs. #38 dynamic) using Games-Howell post-hoc; and for each time points of dynamic samples the cumulative ion release was compared between all groups and the “control” group which released the most amount of ions (#38) using Dunnets post hoc. In Figure 6, the mean cumulative ion concentration at each time point, for each ion, were compared between different sized samples using Games-Howell post-hoc. In Figure 7, for each time point, the cumulative ion release was compared between all group means and the “control” group which released the most amount of ions (#34) using Games-Howell. In Figure 8 the viability of each treated cell group was compared: to the mean values for control (untreated) using Dunnets (Figure 8B), or between all groups and control (untreated) using Games-Howell post hoc. In Figure 9 total protein and ALP were each compared between all treatment groups and control (untreated) means, for each cell line and condition, using Games-Howell post hoc. Statistical analysis was run on groups with samples sizes of: 3 per group for ICP-OES testing; 6 per group for viability testing; and 4 per group for ALP and protein levels.
- Figure 5 should be checked additionally. Were ion concentration measured in static and dynamic mode for each sample? Are all the lines presented in the chart or only 'static' (or dynamic) data? Are significance marks placed correctly (e.g., where are § signs shown in the legend?) ? The standard deviations should be shown as whiskers.
Thank you for the comment, we have added this information to the manuscript to better inform readers in the following section (Results section, Figure 5 notes).
Figure 5. Effect of media replenishment on ion release profiles, for a broad formulation range. Ion release profile was measured in dynamic (lines) and static (single data points at 24 hours) mode only for samples #29, #34, #38, #39, and control for (A) calcium, (B) phosphorus, and (C) silicon. These samples were selected to represent the range of compositions and viability. Single data points at 24 hours represent “static” data in accordance with ISO 10993 (media was collected only once after 24 hours immersion). Sample dimensions were 8.9 mm × 4.5 mm, cured to the initial setting time before immersion. **, and *, § indicates p< 0.01, and 0.05, respectively, for comparisons made between the cumulative release of each respective static or dynamic group at 1 day (*, Games-Howell); and between all groups and the highest eluting sample (#38, §, Dunnets post hoc).
The standard deviation was added to the chart.
Thank you for the warning, there must have been an error while editing. Now the error is corrected and the meaning tags are placed correctly.
- There is no figure legend for Figure S3_2 (to judge from the file name). There is no figure for S2.2 figure legend. I suggest it is a typo in file name?
Thank you for the warning. It was just a typo in the file name. The correct file name is Figure S2.2.
- Dental stem cells are known for their high osteogenic potential and high ALP level. Is it possible to add in Figure 9b the ALP level at the beginning of the experiment. The difference in Dex concentration for 3T3 and MDPC23 should be explained. Fibroblasts, MSC and dental stem cells are very sensitive to Dex concentration. High doses of Dex can induce adipogenic differentiation in some of cell lines of mesenchymal and epithelial-mesenchymal origin.
Thank you for your comment.
The 3-day time point is included to demonstrate the basal rate of ALP expression “early” in the experiment because the MC3T3 cell lines differentiate after 5-7 days.
After considering multiple reviewers’ comments, regarding the use of Dex, we have chosen to remove data involving dex from this manuscript. The authors feel that the Dex data adds little useful information, and confuses/distracts from the main message of the paper. Figure 9 now only includes ALP/BCA data for typical osteogenic media containing ascorbic acid and beta-glycerol phosphate.
Reviewer 2 Report
Dear authors,
I will comment on your paper although being just a biologist, I would disclaim any tissue engineering knowledge. Therefore, I can tell the set up, the aim and the execution of the study appear to be clear and straightforward to me. From the cell culture and especially the cell differentiation aspect I have some questions.
Firstly I would advise to get some better microscopy images. The white balance is not done and the magnification is so poor that cells cannot be distiguished. Also in the materials part the microscopy discription is missing. Did you use phase contrast?
Then the whole osteoblastic medium part is puzzeling. I do not think you investigate the role of dexametasone, so why differentiate? Even if this is of interest in that case you should use other read out as well. Mineralization assays like alizarin red staining would perhaps be interfered by your material, but in that case expression and/or production of BMPs, collagen matrix etc. would be needed. This is why I suggest stick to one osteoblastic medium.
Author Response
Response to the reviewers’ comments
Dear Editor,
We have received your opinion and reviewers’ reports on our manuscript. We have appreciated the deep insight and effort of the reviewers and would like to thank them for all comments and recommendations which we have seriously considered in the revised version. A number of these comments were particularly relevant and helped to improve the original manuscript. We are very grateful to all reviewers for their revisions of our manuscripts. In what follows, we provide answers to the comments and a detailed explanation of how the reviewers’ recommendations have been reflected in the revised text. We have used the following formatting style to distinguish between the reviewers’ questions, our answers, and changes made in the main text: (1) Reviewer´s summarizing text is in italics, (2) questions and comments are in bold font while (3) our responses are in normal plain font and (4) all changes and modifications made in the text of the manuscript are yellow-colored.
Review 2:
Comments and Suggestions for Authors
Dear authors,
I will comment on your paper although being just a biologist, I would disclaim any tissue engineering knowledge. Therefore, I can tell the set up, the aim and the execution of the study appear to be clear and straightforward to me. From the cell culture and especially the cell differentiation aspect I have some questions.
Firstly I would advise to get some better microscopy images. The white balance is not done and the magnification is so poor that cells cannot be distiguished. Also in the materials part the microscopy discription is missing. Did you use phase contrast?
Thank you for your comments. We have added this information to the manuscript to better inform readers. (Material and Methods – Cell culture, page 5) We edit the picture and did a white balance with photos.
The mouse odontoblast-like cell line (MDPC 23, a gift from Jacques Nor at University of Michigan) and mouse fibroblast (L929, American type culture collection, ATCC), were cultured in Dulbecco’s modified Eagles medium/F12 (Gibco, NY, USA) supplemented with 10% fetal bovine serum (FBS, HyClone, USA) and 1% penicillin/streptomycin (SigmaAldrich, St. Louis, USA), at 37°C in 5% CO2 atmosphere. The pre-osteoblast cell line (MC3T3 E14, ATCC CRL2594, LGC Standards GmbH, Germany), was expanded in alpha modified Eagles medium (αMEM Gibco, USA), lacking ascorbic acid and supplemented with 10% fetal bovine serum (FBS, HyClone) and 1% penicillin/streptomycin (Sigma-Aldrich, St. Louis, USA) at 37°C in 5% CO2 atmosphere. For differentiation studies, MC3T3 were cultured in αMEM containing ascorbic acid (αMEM Hyclone, USA). The cell morphology of the morphology was observed by an inverse phase-contrast microscope (Olympus) at 10x magnification.
Thank you for your recommendation. We edit the picture and did a white balance with mentioned photos.
For our experiment, we used phase contrast.
Then the whole osteoblastic medium part is puzzeling. I do not think you investigate the role of dexametasone, so why differentiate? Even if this is of interest in that case you should use other read out as well. Mineralization assays like alizarin red staining would perhaps be interfered by your material, but in that case expression and/or production of BMPs, collagen matrix etc. would be needed. This is why I suggest stick to one osteoblastic medium.
Thank you for your comments. After considering multiple reviewers’ comments, regarding the use of Dex, we have chosen to remove data involving dex from this manuscript. The authors feel that the Dex data adds little useful information, and confuses/distracts from the main message of the paper. Figure 9 now only includes ALP/BCA data for typical osteogenic media containing ascorbic acid and beta-glycerol phosphate.
Reviewer 3 Report
In this work, the authors studied class of bone tissue adhesive PMCs and their ions release profiles. Interestingly, PMCs can stimulate higher ALP activity in osteogenic cells, bridging the in vitro and in vivo and also showing promise for in vivo study. Most parts of this manuscript were well organized. Thus, I would like to recommend publication of this timely work after the authors address the following issues.
- Please remove words such as “the first time…” “novel”… in the manuscript.
- For all ion release profiles, an error bar should be provided.
- Why the author select 70% as the standard for in vitro biocompatibility. Do some literatures support this standard?
Author Response
Response to the reviewers’ comments
Dear Editor,
We have received your opinion and reviewers’ reports on our manuscript. We have appreciated the deep insight and effort of the reviewers and would like to thank them for all comments and recommendations which we have seriously considered in the revised version. A number of these comments were particularly relevant and helped to improve the original manuscript. We are very grateful to all reviewers for their revisions of our manuscripts. In what follows, we provide answers to the comments and a detailed explanation of how the reviewers’ recommendations have been reflected in the revised text. We have used the following formatting style to distinguish between the reviewers’ questions, our answers, and changes made in the main text: (1) Reviewer´s summarizing text is in italics, (2) questions and comments are in bold font while (3) our responses are in normal plain font and (4) all changes and modifications made in the text of the manuscript are yellow-colored.
Review 3:
Comments and Suggestions for Authors
In this work, the authors studied class of bone tissue adhesive PMCs and their ions release profiles. Interestingly, PMCs can stimulate higher ALP activity in osteogenic cells, bridging the in vitro and in vivo and also showing promise for in vivo study. Most parts of this manuscript were well organized. Thus, I would like to recommend publication of this timely work after the authors address the following issues.
- Please remove words such as “the first time…” “novel”… in the manuscript.
Thank you for the comments. We removed the suggestion words in the manuscript.:
We studied a class of bone tissue adhesives (PMCs). A wide range of formulations of PMCs is cytocompatible when in indirect contact with fibroblasts and in direct contact with osteoblasts and odontoblasts.
We have recently shown that a single formulation of a bone tissue adhesive, phosphoserine modified cement (PMC), is safe and resorbable in vivo.
We have also shown that PMCs can stimulate osteogenic protein activity (ALP) and that the amount of phosphoserine and ions that are released are sufficient to explain this phenomenon.
- For all ion release profiles, an error bar should be provided.
Thank you for your recommendation. The standard deviation was added to the all chart with ion release profile.
- Why the author select 70% as the standard for in vitro biocompatibility. Do some literatures support this standard?
Our experiments were based on the standard protocol of ISO 10993-5 where is reported: "Reduction of cell viability by more than 30 % is considered a cytotoxic effect."
Round 2
Reviewer 1 Report
The manuscript has been thoroughly revised and improved. The revised version meets the journal requirements. All my comments were answered and taken into account. I recommend the revised version for publication.